# Interferon Gamma Targeted Therapy: Is It Justified in Primary Sjögren’s Syndrome?

**DOI:** 10.3390/jcm11185405

**Published:** 2022-09-14

**Authors:** Agata Sebastian, Marta Madej, Paweł Gajdanowicz, Maciej Sebastian, Anna Łuczak, Magdalena Zemelka-Wiącek, Marek Jutel, Piotr Wiland

**Affiliations:** 1Department of Rheumatology and Internal Medicine, Wroclaw Medical University, 50-556 Wroclaw, Poland; 2Department of Clinical Immunology, Wroclaw Medical University, 50-368 Wroclaw, Poland; 3Department of General, Minimally Invasive and Endocrine Surgery, Wroclaw Medical University, 50-556 Wroclaw, Poland

**Keywords:** primary Sjögren syndrome, interferon γ, cytokines

## Abstract

Background: The pathomechanism of primary Sjögren syndrome (pSS) is multifactorial. Many cytokines take part in this process, including interferon. The study aimed to quantify certain cytokines involved in the pathomechanism of primary Sjögren syndrome (IL2, IL5, IL6, IL10, IL13, TNFα, IFNγ) and determine their common clinical correlation. On this basis, we discuss the potential use of anti-cytokine drugs in pSS therapy. Methods: The study group consisted of adult patients with a confirmed diagnosis of pSS. Results: The most frequently detected cytokines were IFNγ (82% of patients), TNFα (70%), IL6 (50%), and IL2 (42.5%). In all patients, except for one patient, IFNγ was found in the presence of other specific cytokines. There was no difference in clinical symptoms, age, and laboratory test results between the group of patients with IL-6 + TNFα + IFNγ positive cytokine, and the group of patients in whom they were not detected. There was no correlation between the presence of IL5, IL13, IL2, IL6, IL10, TNFα and musculoskeletal symptoms, skin lesions, glandular domains, pulmonary neurological, lymphadenopathy, biological and hematological domains in ESSDAI (*p* > 0.05). Conclusions: IFNγ most likely plays a central role in the pathomechanism of the disease. We have not noticed a clinical correlation between the three most common cytokines (IL6, IFNγ and TNFα), preliminary research results open up the possibility of searching for new treatments for pSS. The lower percentage of patients with detectable levels of TNFα and IL6 may explain the ineffectiveness of drugs targeting cytokines in clinical trials to date.

## 1. Introduction

Primary Sjögren’s syndrome (pSS) is a chronic systemic disease of connective tissue. Many organs and systems may be involved in the course of the disease. Organ localization determines the further course of the disease [1]. The most common manifestation of pSS includes symptoms of dry mouth and eyes, fatigue, enlargement of large salivary glands, and musculoskeletal symptoms [2]. A characteristic feature of pSS is the production of anti-SSA antibodies and the formation of lymphocytic infiltrates in the affected tissues [3].

The pathomechanism of pSS is similar to that of other autoimmune diseases and is multifactorial (genetic predisposition, environmental trigger, initiation and maintenance of the autoimmune process). Many cytokines take part in this process, including interferon [4]. pSS was initially considered an autoimmune disease in which Th1 cells predominate. However, gradually it began to be observed that both Th1 and Th2 cells are involved in the pathogenesis of the disease depending on its stage. Determining the specific role of the Th1 lymphocyte subpopulation in the progression of pSS is one of the main research goals to help us better understand the disease [5]. Th1 cells produce the pro-inflammatory cytokines interferon-gamma (IFNγ) and tumor necrosis factor-alpha (TNFα). Both of these cytokines regulate the cell-mediated immune response and activate macrophages, NK cells and CD8 + T cells involved in the pSS pathomechanism [6]. Th2 cells participate in the humoral response and the regulation of B lymphocytes [7].

B-cell hyperactivation and the production of specific autoantibodies are observed in pSS. The mechanism supporting this reaction involves cytokines secreted by Th2 cells [8]. The most important cytokines produced by Th2 cells are IL4, IL5, and IL13, but they can also produce IL9, IL10, IL25, and amphiregulin [9]. To date, many studies have been carried out to identify cells essential for the pathogenesis of pSS. Understanding the individual contributions of all interacting cell types and their interactions is of paramount importance in understanding disease pathogenesis and developing effective therapeutic approaches. Most published research papers usually analyze single cells and cytokines in the context of disease pathogenesis. Only in the literature review publications can we find a summary of the entire cytokine pathways involved in the pathomechanism of pSS; therefore, there are still difficulties in the clinical diagnosis of pSS.

The study aimed to quantify certain cytokines involved in the pathomechanism of primary Sjögren syndrome (IL2, IL5, IL6, IL10, IL13, TNFα, IFNγ) and determine their common clinical correlation.

On this basis, we discuss the potential use of anti-cytokine drugs in pSS therapy.

## 2. Materials and Methods

### 2.1. Study Group

The study group consisted of patients with a confirmed diagnosis of pSS based on the current ACR-EULAR Classification Criteria of 2016 [3], hospitalized at the Rheumatology Clinic. The inclusion criteria for the study were the age of over 18 years on the day of the study, diagnosis of pSS [3], no associated connective tissue diseases, access to the patient’s complete medical records and obtaining informed consent to conduct the study from each participant.

The exclusion criteria included lymphoproliferative diseases, including lymphoma, sarcoidosis, hepatitis A, B or C, autoimmune liver disease, HIV infection, other connective tissue diseases (including rheumatoid arthritis, systemic lupus erythematosus), asthma, allergy, past head and neck radiation treatment, graft versus host disease, past transplantation, past blood transfusion for any reason, infectious disease or viral infection in the last three months before the study. None of the patients used non-steroidal anti-inflammatory drugs 72 h before blood sampling for laboratory tests. After obtaining the written consent of all patients, a detailed history was taken, a physical examination was performed, and blood was collected for laboratory determinations. Based on the collected data, disease activity was assessed using the EULAR Disease Activity Index (ESSDAI) [10] and the severity of dryness, fatigue and pain using the EULAR Sjögren’s Syndrome Patient Reported Index (ESSPRI) [11].

### 2.2. Serum Sampling

The standard venipuncture method collected 10 milliliters of venous blood from the elbow fossa. One part of the serum was separated from the blood by centrifugation, then stored at −80 °C until further biochemical analyses (IL5, IL13, IL2, IL6, IL10, TNFα, IFNγ) and thawed immediately prior to testing. Serum cytokine levels were assessed using LEGENDplex kits (BioLegend, San Diego, CA, USA). According to the manufacturer’s instructions, a BD CANTO2 flow cytometer and LEGENDplex data analysis software were used for the analysis. The minimum detection limits for individual cytokines were 36.57 pg/mL for IFNγ, 8.33 pg/mL for IL5, 8.98 pg/mL for IL13, 8.83 pg/mL for IL2, 8.03 pg/mL for IL6, 8.78 pg/mL for IL10, 7.63 pg/mL for TNFα.

Antinuclear antibodies (ANA) were determined in the second part of the serum and quantified using the technique of indirect immunofluorescence (on Hep2 cells). A titer of 1:320 or higher was considered positive. Anti-Ro/SSA and anti-La/SSB autoantibodies were detected using an ELISA (enzyme-linked immunosorbent assay) to quantify IgG autoantibodies (EUROIMMUN). Quantitative determination of rheumatoid factor (RF) in blood serum was performed using an immunonepheloetry test. In addition, all patients underwent tests: peripheral blood count, C-reactive protein (CRP), erythrocyte sedimentation index (ESR), electrolytes, cryoglobulins, complement components C3 and C4, electrophoresis of serum proteins, and the concentration of IgG and IgM immunoglobulins. Each patient also had a urine sample analyzed for abnormalities in its composition.

### 2.3. Required Consents

The study was conducted in accordance with the principles of the Helsinki Declaration and was approved by the local Bioethics Committee (Decision No. KB-114/2021). Written informed consent was obtained from all participants. The study was carried out with funds allocated as part of a subsidy for the maintenance and development of the research potential of the Medical University of Wroclaw, Poland (SUB.A270.21.032).

### 2.4. Statistical Analysis

The analysis was performed using the Statistica 10 software package (JMP Statistical Discovery, Cary, NC, USA).

The Mann–Whitney U test was used to compare the distributions of quantitative variables in two independent groups. The chi-square test was used to verify the relationship between dichotomous variables. Significance was found at *p* ≤ 0.05.

## 3. Results

The study group consisted of 40 pSS patients (37 women and three men) with a mean age of 48 years. More than 80% of patients had dryness. The patients had moderately active disease with a mean ESSDAI of 9.0 points and ESSADI between 3 and 5 points in the ESSPRI domains. A total of 90% of patients had a positive rheumatoid factor. The demographic data of pSS patients are shown in Table 1. A total of 17% of patients were not using disease-modifying drugs; half had hydroxychloroquine as baseline therapy. A total of 27% of patients required combined treatment with glucocorticosteroids due to organ changes (Table 2). Before the blood sample was collected for testing, all patients were on a stable dose of drugs for a minimum of three months.

The most frequently detected cytokines in the studied population were IFNγ (82% of patients), TNFα (70%), IL6 (50%), and IL2 (42.5%). IL13 (32.5%), IL5 (20%), and IL10 (7.5%) were recorded in a smaller percentage of patients. Concentrations higher at least twice the normal range value were found in 88% of patients for IFNγ, 65% for IL6, 53% for IL2, and 50% for TNFα. In all patients, except for one patient (32 out of 33 subjects), IFNγ was observed always with the presence of other specific cytokines (IL5 or/and IL13 or/and IL2 or/and IL6 or/and IL10 or/and TNFα). The exact distribution of the individual cytokines is shown in Table 3.

Considering the three most common cytokines, IL6, TNFα and IFNγ, there was no difference in clinical symptoms, age, and laboratory test results between the group of patients with these three cytokines (IL6 + TNFα+ IFNγ +), and the group of patients in whom they were not detected (IL6- TNFα- IFNγ-) (Table 4). There was no correlation between the presence of IL5, IL13, IL2, IL6, IL10, TNFα and musculoskeletal symptoms, skin lesions, glandular domains, pulmonary, neurological, lymphadenopathy, biological and hematological domains in ESSDAI (*p* > 0.05).

There were no statistically significant correlations between other individual cytokines (*p* > 0.05).

## 4. Discussion

In our study, the most frequently detected cytokines among pSS patients were IL6, TNFα and IFNγ. Most of them were values greater than the minimum double detection for a given molecule. In the case of a significantly smaller percentage of tested sera from pSS patients, the presence of IL2, and IL13 was found, and, least frequently, that of IL5 and IL10. In most patients (32 out of 33) in whom IL5 or/and IL13 and/or IL2 or/and IL6 or/and IL10 and/or TNFα were detected, IFNγ was also detected in increased concentration. We showed no clinical correlation between the increased levels of TNFα, IFNγ and IL6 and the clinical picture of the disease. The study groups—positive for TNFα, IFNγ and IL6 and the negative group for these three cytokines—did not differ in terms of the analyzed clinical features and the course of the disease.

After IFNγ, the second most common cytokine in pSS patients is TNFα. It is one of the critical elements involved in the pathogenesis of pSS. In animal models of SS without autoantibodies, it has been shown that the overexpression of TNFα induces symptoms of exocrine gland inflammation [12]. On the other hand, we know that TNFα inhibitors are successfully used in the treatment of arthritis in the course of other diseases, e.g., rheumatoid arthritis, and joint symptoms are one of the more common symptoms in the course of pSS [13]. The data so far indicate also the multidirectional pro-inflammatory effect of cytokines involved in the pathogenesis of arthritis [14]. Hence, it would seem that the use of drugs that block this cytokine should benefit the treatment of pSS. However, the use of etanercept and infliximab in clinical trials did not bring the expected results [15,16,17]. The relationship between TNFα and the development of pSS remains unclear. It seems that the concentration of this cytokine may play a role in pSS. In our observation, only half of the patients with detectable TNFα had concentrations above twice the value for its detection. The values of IFNγ concentrations were different, where a minimum of double the detection value or a higher concentration was observed in almost 90% of the examined patients. Through IFNγ, the expression of BAFF (B-cell activating factor, tumor necrosis factor ligand superfamily member 13B) can be stimulated, which in turn influences the production of B cells and the production of autoantibodies [18].

What is interesting in our results is that, regardless of the percentage distribution of the determining factors and interleukin concentrations, elevated IFNγ values were found in all patients. As shown in our recently published study, the group of pSS patients with detectable IFNγ differed in terms of the clinical picture from patients without detected IFN, younger age, and higher values of RF and ESSDAI [19]. Hence, it seems that the modification of the IFN mechanism of action should be more effective in the treatment of pSS patients. However, this requires additional, well-designed clinical trials with the use of new biological therapies, such as, for example, Janus kinase inhibitors (JAK) [20]. Recent studies indicate the influence of epigenetic changes on DNA methylation/hydroxymethylation processes in pSS, mainly on genes regulated by two cytokines-IFNα and IFNγ, as well as through oxidative stress pathways. The JAK-STAT system (JAK kinase—signal transducer and activator of transcription (STAT)) is activated by IFN and reactive oxygen species (ROS) [21]. The results of a prospective, randomized, double-blind, multicenter clinical study showed that tofacitinib (a JAK kinase inhibitor), used, among other things, in the treatment of rheumatoid arthritis and psoriatic arthritis, has a positive effect on the reduction of dry eye symptoms in pSS patients [22]. Another JAK kinase inhibitor—baricitinib—inhibits JAK-STAT signaling, IFNγ-induced expression of the CXCL10 molecule, and cell chemotaxis, thus becoming another promising drug in pSS therapy [23].

Another interleukin whose blocking could theoretically inhibit the development of pSS is IL6. Their elevated values were found in half of our patients, of which 65% of them were at least twice as high. In studies on mouse models, it was noticed that IL6 induces the differentiation of salivary gland epithelial cells into Tfh cells, and the use of IL6 inhibitors can significantly improve the course of SS [24]. However, no significant benefit of IL6 receptor blockade in humans has been demonstrated. In studies conducted so far with tocilizumab (an IgG1 monoclonal antibody that specifically binds to IL6 receptors, both soluble and membrane-bound-sIL-6R and mIL-6R), the planned endpoints initially have not been achieved. On the other hand, some improvement was observed in patients with joint involvement [25,26]. However, it seems that IL6 is not the primary factor in pSS B-cell hyperresponsiveness.

IL2 promoting the proliferation of Treg cells leads to clinical improvement in pSS. Miao et al. demonstrated that low-dose IL-2 treatment increased CD4Treg cell count compared to the baseline. On the other hand, the Th17/Treg cell ratio is reduced. As demonstrated by the authors, the absolute number of CD4Treg cells was lower in the case of higher pSS activity [27]. These results are consistent with those presented by us. In our group of patients, less than half of the respondents had detectable IL2, which may be consistent with recent suggestions that this cytokine is considered more of an immunomodulatory rather than immunosuppressive activity. Additionally, Luo et al. observed decreased IL2 values and increased IL6 in pSS patients [28], which is consistent with our observations.

As in other publications [29], we observed IL5 concentrations below the detection threshold in our patients, and it seems that this is a mechanism of less importance in the development and maintenance of the autoimmune process in pSS. The distribution for IL10 concentrations, which was the least frequently observed interleukin among the studied patients, was similar. Its role in the pathomechanism of the pSS remains unknown [27].

In recent years, the role of IL13 in autoimmune and allergic diseases has been emphasized. IL13 can regulate several subtypes of T-helper (Th) cells and influence their transformation, including Th1, Th2, and T17. Therefore, it is suspected that it may play an important role in the regulatory mechanisms of the immune system. In the studies carried out so far on murine models and based on histopathological examinations of the salivary glands of patients diagnosed with pSS, an increased concentration of IL13 was found [30], which we did not confirm in our study, analyzing the concentration of cytokines in the blood serum. IL13 in our pSS group was one of the lowest percentages of all cytokines determined. It is possible that the balance between Th1/Th2 cells changes with the progression of immune disorders in pSS, and IL13 plays a potentially crucial local role in regulating glandular function.

The limitation of our study is the relatively small number of patients. In addition, the recruited patients were patients with active pSS (ESSDAI > 5 points) and partially treated with disease-modifying drugs, which could affect the cytokine level. The study also did not consider the division into the disease phenotype: patients with only symptoms of dryness vs. patients with organ involvement.

## 5. Conclusions

The results of our studies indicate that IFNγ most likely plays the central role in the pathomechanism of the disease and could be the target in the precision medicine era. These preliminary research results open up the possibility of searching for new treatments for pSS. A lower percentage of patients with detectable levels of TNFα and IL6 may explain the ineffectiveness of drugs targeting cytokines in clinical trials to date. A more detailed understanding of the pathogenetic pathways of the disease is needed, which could result in the definition of new therapeutic targets for biological drugs.

## Figures and Tables

**Table 1 jcm-11-05405-t001:** Baseline characteristics of the study pSS patients.

Variables, Units	
Age, years (min-max)	48 (21–68)
Female N (%)	37 (95)
Time after pSS diagnosis, years (min-max)	4 (1–28)
Dry eye symptoms N (%)	34 (85)
Dry mouth symptoms N (%)	36 (90)
ESSPRI mean (min-max):	
- dryness	4.9 (0–10)
- fatigue	5.1 (0–10)
- pain	3.1 (0–10)
ESSDAI mean (min-max)	9.0 (1–21)
Domains > 0 points N (%):	
- constitutional	1 (2)
- lymphadenopathy	5 (12)
- glandular	9 (22)
- articular	20 (50)
- cutaneous	4 (10)
- pulmonary	8 (20)
- renal	1 (2)
- muscular	0 (0)
- peripheral nervous system	0 (0)
- central nervous system	0 (0)
- haematological	10 (20)
- biological	20 (50)
ANA ≥ 1:320 N (%)	40 (100)
Anti-SSA ab N (%)	38 (95)
Anti-SSB ab N (%)	26 (65)
Focus score ≥ 1 N (%)	38 (95)
ESR (nv 0–25 mm/hr) mean (min-max)	17 (2–68)
CRP (nv 0–5 mg/l) mean (min-max)	1.7 (0.2–13)
RF (nv 0–30 IU/mL) mean (min-max)	35 (5–411)
C3- hypocomplementemia (nv < 0.83 g/l) N (%)	2 (5)
C4- hypocomplementemia (nv < 0.15 g/l) N (%)	8 (20)
Chronic diseases N (%):	
- hypertension	5 (12)
- hypercholesterolemia	4 (10)
- thyroid nodules	1 (2.5)

Abbreviations: N—number of patients; %—percent of patients; ANA—antinuclearantibodies; ab—antibodies; focus score—the number of mononuclear cell infiltrates containing at least 50 inflammatory cells in a 4 mm^2^ glandular section; ESSPRI—EULAR Sjögren’s Syndrome Patient Reported Index; ESSDAI—EULAR Sjögren’s Syndrome Disease Activity Index; nv—normal value; RF—rheumatoid factor; CRP—C-reactive protein; ESR—erythrocyte sedimentation rate.

**Table 2 jcm-11-05405-t002:** Baseline characteristics of the study cohort–treatment.

Treatment (Dose)	N (%)
Hydroxychloroquine (200 mg/day)	20 (50)
Methotrexate (10–25 mg/week)	5 (12)
Azathioprine (100–175 mg/day)	5 (12)
Cyclosporine (50–150 mg/day)	2 (5)
Mycophenolate mofetil (1–2 g/day)	1 (2)
Prednisone (≤5 mg/day)	6 (15)
Combination therapy	11 (27)
No therapy	7 (17)

Abbreviations: N—number of patients; %—percent of patients.

**Table 3 jcm-11-05405-t003:** Distribution of the cytokine profile in the serum of pSS patients.

Type of Molecule	Number of Patients n/%	Detection Value > 2 nv n/%	Patients with Positive IFNγ (n)	Patients with Negative IFN γ (n)
IL-5	8/20	3/37.5	8	0
IL-13	13/32.5	4/31	12	1
IL-2	17/42.5	9/53	17	0
IL-6	20/50	13/65	20	0
IL-10	3/7.5	1/33	3	0
TNFα	28/70	14/50	28	0
IFNγ	33/82.5	29/88	NA	NA

Abbreviation: n—number of patients; %—percent of patients; IL-interleukin; TNFα—tumor necrosis factor alpha; IFNγ—interferon gamma; NA—not applicable.

**Table 4 jcm-11-05405-t004:** Comparison of pSS patients with IL6, TNFα and IFNγ presence demonstrated with a group of negative patients in terms of their determination.

	Positive for IL6 and TNFα and IFNγ	Negative for: IL6 and TNFα and IFN γ	*p*-Value
Age (years)	44.3	51.5	0.1
Age at pSS diagnosis (years)	41.9	46.5	0.12
Mean ESSDAI	8.8	10.5	0.2
Mean ESSPRI:			
- pain	2.9	3.4	>0.05
- fatigue	4.7	5.4	>0.05
- dryness	4.2	5.4	>0.05
FS ≥1 in LSGB	33% of patients	45% of patients	>0.05
Mean RF (nv < 14 IU/mL)	30.9	39.1	>0.05

Abbreviation: IL—interleukin; TNFα—tumor necrosis factor-alpha; IFNγ—interferon-gamma; ESSDAI—the EULAR Disease Activity Index; ESSPRI—the EULAR Sjögren’s Syndrome Patient Reported Index; FS—focus score; LSGB—labial salivary glands biopsy; RF— rheumatoid factor; pSS—primary Sjögren’s syndrome.

## Data Availability

The datasets used and/or analyzed during the current study are available from the corresponding author upon reasonable request.

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
