# Peer review of "Interferon Gamma Targeted Therapy: Is It Justified in Primary Sjögren’s Syndrome?"

_jcm, 2022, doi:10.3390/jcm11185405_

Round 1

Reviewer 1 Report

This study proposes the quantification of certain cytokines involved in the pathomechanism of Sjogren Syndrome (IL5, IL13, IL 2, IL 6, IL 10, TNFα, IFNγ) and determine their common clinical correlation.

Although fairly interesting, there are many issues that must be addressed before publication.

Please see enclosed PDF for further details

Author Response

Dear Reviewer,

Thank You very much for the time You spent reviewing the manuscript we have proposed.

We tried to correct the manuscript with all suggestions- please see the pdf for details and the main manuscript. Additionally, the text has been verified and updated by the native speaker.

We hope that the article will now be suitable for publication. If You have any other suggestions, please let us know.

Sincerely,

Authors  

Reviewer 2 Report

In the present paper the authors evaluate the distribution of cytokines involved in the pathomechanism of Sjogren’s Syndrome. The data although preliminary are interesting, but the approach and analysis of data are rather shallow.

1. The authors explore the presence of specific cytokines in the blood of patients. It would also be advisable to investigate more specific tissues, in particular the exocrine glands, in order to have a more complete view.

2. It would be informative to display in a table all demographic,  clinical and laboratory features of pSS patients.

3. The data presented by the authors are exclusively qualitative. It would also be interesting to present a quantitative overview of the cytokines detected and analyze a possible correlation between them.

4. Although the authors state that “None of the patients used non-steroidal anti-inflammatory drugs 72 hours before blood sampling for laboratory tests”,  long-term drug treatments and comorbidities may well influence the cytokines production. On this basis, authors should present the comorbidities and treatments in pSS patients studied.

5. The main purpose of the study is not well defined. In my opinion, the title and conclusions are beyond the main purpose.

Author Response

Dear Reviewer,

Thank You very much for the time You spent reviewing the manuscript we have proposed.

We tried to correct the manuscript with all suggestions. Below we will send a detailed answer to the questions. All amendments are marked in yellow in the text. Additionally, the text has been verified and corrected by the native speaker.

We hope that the article will now be suitable for publication. If You have any other suggestions, please let us know.

Sincerely,

Authors  

  1. The authors explore the presence of specific cytokines in the blood of patients. It would also be advisable to investigate more specific tissues, in particular the exocrine glands, in order to have a more complete view.

Thank You very much for this suggestion. The study of the salivary glands was not the aim of our work. The concept is interesting but requires further research. We had limited funds for this study. Tissue assays were associated with the use of another expensive assay method. So one of the major key limitations was the lack of funding that would also allow the determination of the specific distribution of cytokines in the tissues.

2. It would be informative to display in a table all demographic,clinical and laboratory features of pSS patients.

* We have added Table 1 and 2 with the demographic data of the study group.

3. The data presented by the authors are exclusively qualitative. It would also be interesting to present a quantitative overview of the cytokines detected and analyze a possible correlation between them.

We already analysed the correlation between cytokines at the initial stage of analysing the results. However, due to the lack of statistical value of these data, we did not present them all in the text to make the manuscript more readable. We have not observed any statistically significant correlations between individual cytokines, which we have completed in the text.

4. Although the authors state that “None of the patients used non-steroidal anti-inflammatory drugs 72 hours before blood sampling for laboratory tests”,long-term drug treatments and comorbidities may well influence the cytokines production. On this basis, authors should present the comorbidities and treatments in pSS patients studied.

* Thank You for this valuable comment.  We have added Table 1 and 2 with the demographic data of the study group.

5. The main purpose of the studyis not well defined.In my opinion, the title and conclusions are beyond the main purpose.

* Thank You for this valuable suggestion. We reformulated the aim, discussion and conclusion again.

Round 2

Reviewer 1 Report

The manuscript has been improved

Reviewer 2 Report

The authors supplemented the paper with the required suggestions or highlighted the study's lack in limitations